# The Effects of Niobium and Molybdenum on the Microstructures and Corrosion Properties of CrFeCoNiNbxMoy Alloys

**DOI:** 10.3390/ma15062262

**Published:** 2022-03-18

**Authors:** Chun-Huei Tsau, Yi-Hsuan Chen, Meng-Chi Tsai

**Affiliations:** Institute of Nanomaterials, Chinese Culture University, Taipei 111, Taiwan; psfohs1130@gmail.com (Y.-H.C.); asd99586@yahoo.com.tw (M.-C.T.)

**Keywords:** high-entropy alloy, CrFeCoNiNb_x_Mo_x_, CrFeCoNiNb_x_Mo_1−x_, microstructure, hardness, corrosion

## Abstract

The present work systematically investigated the effects of niobium and molybdenum on the microstructures and corrosion properties of high-entropy CrFeCoNiNb_x_Mo_x_ and CrFeCoNiNb_x_Mo_1−x_ alloys, the maximum content of (Nb + Mo) was 20 at.%. All of the alloys were prepared by arc melting under an argon atmosphere. In CrFeCoNiNb_x_Mo_x_ alloys (x = 0.15, 0.3 and 0.5), increasing Nb and Mo content would change the microstructure of the alloy from a hypoeutectic structure (x ≤ 0.3) to a hypereutectic one (x = 0.5). All of the CrFeCoNiNbxMo_1−x_ alloys (x = 0.25, 0.5 and 0.75) had a hypereutectic microstructure. Only two phases were analyzed in these alloys, which were face-centered cubic (FCC) and hexagonal close packing (HCP). Increasing the content of Nb and Mo increases the hardness of the alloys by the effects of the solid solution strengthening and formation of the HCP phase. The potentiodynamic polarization curves of these alloys were also measured in 1 M sulfuric acid and 1 M sodium chloride solutions to evaluate the corrosion resistance of these alloys. The CrFeCoNiNb_0.3_Mo_0.3_ alloy had the smallest corrosion rate (0.0732 mm/yr) in 1 M deaerated H_2_SO_4_ solution, and the CrFeCoNiNb_0.15_Mo_0.15_ alloy had the smallest corrosion rate (0.0425 mm/yr) in 1 M deaerated NaCl solution. However, the CrFeCoNiNb_0.5_Mo_0.5_ alloy still had the best combination of corrosion resistance and hardness in the present study.

## 1. Introduction

The alloys used in this study were prepared under the high-entropy alloy concept [1,2,3]. This high-entropy alloy concept provides researchers to develop new materials with suitable properties for applications. Researchers can smartly select the elements to prepare the materials, and many high-entropy alloys (HEAs) were thus produced with excellent mechanical, physical and chemical properties. Moreover, the shapes of high-entropy materials can be bulk alloys, thin films or coating alloys. For example, the casting, homogenization, cold rolling, recrystallization and deformation mechanism of equiatomic CoCrFeMnNi high-entropy alloy were well investigated [4,5]. The microstructures and compression properties of CoCrFeNiTiAl_x_ high-entropy alloys were tested, and results indicated that the CoCrFeNiTiAl alloy had good compressive strength and elastic modulus [6]. The elements with a high melting point were selected to produce the refractory alloys, such as NbMoTaW, VNbMoTaW and HfTaTiNbZr-based alloys [7,8]. The high-entropy alloys can be prepared by mechanical alloying (MA) to obtain the alloys with nanocrystalline structures and enhance the properties [9,10].

Cobalt, chromium and nickel are wildly used to produce alloys with good corrosion resistance. Other elements are selected and added into CoCrNi alloy to change the microstructures and enhance the mechanical properties, such as CoCrFeMnNi alloy [11]. Co–CrFeNiSn has good passivation in sodium chloride solution compared with stainless steels [12]. The AlCoCrFeNiTi_0.5_ coating was fabricated by laser cladding coating and showed the optimal performance of corrosion and mechanical properties [13]. Minor additions of molybdenum could improve the corrosion resistance of the AlCrFe_2_Ni_2_ alloy by suppressing pit formation [14]. The addition of molybdenum could increase the corrosion resistance was observed in the (CoCrFeNi)_100−x_Mo_x_ high-entropy alloys [15]. The study on the Al_0.4_CrFe_1.5_MnNi_0.5_Mo_x_ alloys indicated that adding molybdenum can effectively improve the impedance of passive film and reduce the corrosion current density and thus form a more stable passivation film [16]. The non-equimolar Cr_19_Fe_22_Co_21_Ni_25_Mo_13_ alloy possessed better corrosion resistance compared with 304 stainless steel in both deaerated 1 M HNO_3_ and 1 M HCl solutions [17]. The corrosion resistance of FeCuNbSiB and CrFeCoNiNb_x_ alloys can be improved by adding niobium [18,19].

In our previous study on the corrosion properties of FeCoNi and CrFeCoNi alloys [20], the FeCoNi alloy had a better corrosion resistance by comparing with CrFeCoNi alloy. After adding molybdenum, the corrosion resistance of FeCoNiMo was not as good as that of the CrFeCoNiMo alloy [21]. This indicates that chromium is a very important element in developing a corrosion-resistant alloy. Therefore, the present work studied the effect of adding Nb and Mo on the CrFeCoNi alloy and evaluated the potential of the application.

## 2. Materials and Methods

The CrFeCoNiNb_x_Mo_x_ and CrFeCoNiNb_x_Mo_1−x_ alloys were prepared by arc-melting under argon atmosphere after accurate weighting. Each melt was about 100 g. Table 1 lists the nominal compositions of the alloys. The microstructures of the alloys were examined by a scanning electron microscope (SEM, JEOL JSM-6335, JEOL Ltd., Tokyo, Japan) after regular metallurgical processes. The compositions of the alloys and the phases existing in the alloys were analyzed by an energy dispersive spectrometer (EDS). An X-ray diffractometer (XRD, Rigaku ME510-FM2, Rigaku Ltd., Tokyo, Japan) was used to identify the phases in the alloys, and the scanning rate was fixed at 0.04 degrees per second. A Vicker’s hardness tester (Matsuzawa Seiki MV1, Matsuzawa Ltd., Akita, Japan) was used to measure the hardness of the alloys, and the loading force was 19.61 N (2000 gf).

The potentiodynamic polarization curves of the alloys were tested by a three-electrode electrochemical device (Autolab PGSTAT302N, Metrohm Autolab B.V., Utrecht, The Netherlands). One electrode was the specimens mounted in epoxy resin with an exposed area of 0.196 cm^2^ (0.5 cm in diameter). The second electrode was a counter (a platinum wire). The third electrode was a reference one, which was a saturated silver chloride electrode (Ag/AgCl, SSE). This potential of the reference Ag/AgCl electrode was 0.197 V higher than the standard hydrogen electrode (SHE) at 25 °C [22]. All of the potentiodynamic polarization curves were tested at 30 °C, and the scanning rate of the potentiodynamic polarization test was 1 mV per second. Nitrogen bubbled through the process to degas the oxygen in the solutions. The solutions of 1 M sulfuric acid and 1 M sodium chloric solutions were prepared by reagent-grade acids and deionized water.

## 3. Results and Discussion

This work was divided into two parts to investigate the effect of Nb and Mo content on the effects of CrFeCoNiNb_x_Mo_y_ alloys. Part 1 was the CrFeCoNiNb_x_Mo_x_ alloys, and x was 0.15, 0.3 and 0.5. The same amount of Nb and Mo was added to the CrFeCoNi alloy. The microstructure revolution of CrFeCoNiNb_x_Mo_x_ alloys were studied in this part; the relationships between the properties and the microstructures were also investigated. Part 2 studied the properties of CrFeCoNiNb_x_Mo_1−x_ alloys, where x was 0.25, 0.5 and 0.75. This part studied the effect of different ratios of Nb and Mo on the microstructures and properties of the alloys. The total amount of Nb and Mo was fixed at one part (20 at.%) because the alloys would easily crack during solidification if an excess amount of Nb and Mo was added.

### 3.1. CrFeCoNiNb_x_Mo_x_ Alloys

The microstructures of as-cast CrFeCoNiNb_x_Mo_x_ alloys, x = 0.15, 0.3 and 0.5, are shown in Figure 1. In our previous study, the CrFeCoNi alloy had an FCC-structured granular microstructure and some Cr-rich precipitates with HCP structure [20]. After adding Nb and Mo, the microstructures of as-cast CrFeCoNiNb_x_Mo_x_ alloys became dendritic ones. The dendrites of CrFeCoNiNb_x_Mo_x_ alloys showed a single phase, and the interdendrities of CrFeCoNiNb_x_Mo_x_ alloys showed a eutectic structure. Table 2 shows the chemical compositions of the alloys and the phases existing in the alloys. According to our previous study [23], the HCP phase had higher Nb and Mo content, and FCC had higher Cr- Fe and Ni content. Therefore, the FCC and HCP phases were easy to identify by detecting the compositions. The dendrites of CrFeCoNiNb_0.15_Mo_0.15_ and CrFeCoNiNb_0.3_Mo_0.3_ alloys were an FCC phase and the dendrites of CrFeCoNiNb_0.5_Mo_0.5_ alloy were an HCP-structured laves phase. All of the interdendrites of CrFeCoNiNb_x_Mo_x_ alloys were a eutectic structure with two phases which were FCC and HCP phases (laves phase). This indicated that the alloy changed from a hypoeutectic alloy to a hypereutectic one. That is, CrFeCoNiNb_0.15_Mo_0.15_ and CrFeCoNiNb_0.3_Mo_0.3_ alloys were hypoeutectic alloys, and CrFeCoNiNb_0.5_Mo_0.5_ alloy was a hypereutectic alloy.

Figure 2 shows the XRD patterns of the CrFeCoNiNb_x_Mo_x_ alloys. Two phases of FCC and laves phases (HCP structure) were identified in these alloys. The FCC phase was the main phase in the CrFeCoNiNb_0.15_Mo_0.15_ alloy. Only a small amount of laves phase was in this alloy. Increasing the Nb and Mo content resulted in increasing the amount of laves phase. The laves phase became the major phase in the CrFeCoNiNb_0.5_Mo_0.5_ alloy. This result quite matches with the SEM observation.

Figure 3 plots the hardness of CrFeCoNiNb_x_Mo_x_ alloys as a function of Nb and Mo content. Increasing Nb and Mo content would almost linearly increase the hardness of the CrFeCoNiNb_x_Mo_x_ alloys. The hardness of CrFeCoNiNb_0.15_Mo_0.15_ alloy was only 215 HV, and the hardness of CrFeCoNiNb_0.5_Mo_0.5_ reached 553 HV. Adding Nb and Mo into CrFeCoNi alloy increases the hardness because the atomic radiuses of niobium and molybdenum are larger than those of the other elements. The atomic radiuses of niobium and molybdenum are 1.43 and 1.40 Å, respectively; the atomic radiuses of cobalt, chromium, iron and nickel are 1.253, 1.249, 1.241 and 1.243 Å, respectively [24]. Therefore, the hardness of the CrFeCoNiNb_x_Mo_x_ alloys increased due to the solid solution strengthening effect. Our previous work indicated that increasing niobium and molybdenum content increases the density of dislocation in the FCC phase [23]. This was another effect enhancing the hardness of the alloys. Additionally, the hardness of the HCP-structured laves phase was higher than that of the FCC phase because the slip system of the HCP phase was less than that of the FCC phase. The hardness of CrFeCoNiNb_x_Mo_x_ alloys increased after adding more niobium and molybdenum elements due to the increase in the HCP phase and solid solution strengthening effect.

Figure 4 shows the potentiodynamic polarization curves of the as-cast CrFeCoNiNb_x_Mo_x_ alloys in 1 M deaerated H_2_SO_4_ solution at 30 °C. The potentiodynamic polarization data of these polarization curves are listed in Table 3. The potentiodynamic polarization curves of the alloys with potential negative than corrosion potential (*E*_corr_) was the cathode. The potentiodynamic polarization curves of the alloys with potential positive than corrosion potential was the anode. The corrosion potential (*E*_corr_) of CrFeCoNiNb_x_Mo_x_ alloys was very close. The standard electrode potential of the elements used in present work is listed in Table 4 [25]. The niobium is more active than the other elements because the standard electrode potential of niobium is more negative. Therefore, the corrosion potential of CrFeCoNiNb_0.5_Mo_0.5_ alloy had the most negative corrosion potential (*E*_corr_). The corrosion current densities (*i*_corr_) of CrFeCoNiNb_x_Mo_x_ alloys were all around 10 μA/cm^2^. The potentiodynamic polarization curve of CrFeCoNiNb_0.15_Mo_0.15_ alloy had an apparent anodic peak, and the other alloys had no anodic peak. The passivation potential (*E*_pp_) and critical current density (*i*_crit_) of the anodic peak of CrFeCoNiNb_0.15_Mo_0.15_ alloy is listed in Table 3. Thus, the CrFeCoNiNb_0.3_Mo_0.3_ and CrFeCoNiNb_0.5_Mo_0.5_ alloys easily entered into the passivation regions and formed the passive films during corrosion in H_2_SO_4_ solution. The lowest passivation current densities (*i*_pass_) of these alloys were about 12 A/cm^2^. All the passivation regions of these alloys were breakdown at a potential (*E*_b_) of about 1.2 V (SHE) due to oxygen evolution reaction [26].

Figure 5 displays the potentiodynamic polarization curves of the as-cast CrFeCoNiNb_x_Mo_x_ alloys in 1 M deaerated NaCl solution at 30 °C. The potentiodynamic polarization data of these polarization curves are listed in Table 5. The corrosion potential (*E*_corr_) of CrFeCoNiNb_0.15_Mo_0.15_ alloy was more negative than the other alloys, and CrFeCoNiNb_0.15_Mo_0.15_ alloy also had the smallest corrosion current density (*i*_corr_). All of the potentiodynamic polarization curves of CrFeCoNiNb_x_Mo_x_ alloys in 1 M deaerated NaCl solution had apparent anodic peaks. Additionally, the potentiodynamic polarization curves of Cr–FeCoNiNb_0.15_Mo_0.15_ and CrFeCoNiNb_0.3_Mo_0.3_ alloys had small secondary anodic peaks. The CrFeCoNiNb_0.15_Mo_0.15_ alloy had the lowest passivation current density. The passivation current densities (*i*_pass_) of these alloys were about 9–15 μA/cm^2^.

### 3.2. CrFeCoNiNb_x_Mo_1−x_ Alloys

The microstructures and properties of CrFeCoNiNb_x_Mo_1−x_ alloys were studied in this part. The total amount of Nb and Mo of these alloys was kept as one part; the amount of Nb and Mo thus equaled 20 at.%. The microstructures of as-cast CrFeCoNiNb_x_Mo_1−x_ alloys, x = 0.25 and 0.75, are shown in Figure 6. All of the CrFeCoNiNb_x_Mo_1−x_ alloys, x = 0.25, 0.5 and 0.75, had a hypereutectic microstructure because a large amount of Nb and Mo were added into these alloys. The dendrites of CrFeCoNiNb_x_Mo_1−x_ alloys were an HCP phase (laves phase), and the interdendrites of CrFeCoNiNb_x_Mo_1−x_ alloys were a eutectic structure with FCC and HCP phases (laves phase). Table 6 lists the chemical compositions of the alloys and the phases existing in the alloys. As described above, the Nb and Mo content in the HCP phase of the alloys is higher than that in the FCC phase.

Figure 7 shows the XRD patterns of the CrFeCoNiNb_x_Mo_1−x_ alloys. Two phases existing in these alloys were FCC and HCP-structured laves phases. The HCP phase was the main phase in these CrFeCoNiNb_x_Mo_1−x_ alloys, and the FCC phase became the minor phase in these alloys. Figure 8 displays the hardness of CrFeCoNiNb_x_Mo_1−x_ alloys, some of the data were from our previous study [23]. According to our previous study, the overall hardness of the alloy was contributed by the hard HCP dendrites and the soft interdendrities (HCP + HCC). The hardness of both HCP and FCC phases increased with increasing the content of Nb and Mo due to solid solution strengthening. However, the hardness of the FCC phase was still softer than that of the HCP phase. Because the hardness of interdendrities was softer than that of dendrites, increasing the volume fraction of interdendrite would decrease the overall hardness of the alloy. The CrFeCoNiNb_0.5_Mo_0.5_ alloy had the lowest hardness among these alloys because of the largest ratio of interdendrities in this alloy. The CrFeCoNiNb_0.75_Mo_0.25_ alloy had the highest hardness of 625 HV among the alloys in the present study.

Figure 9 shows the potentiodynamic polarization curves of the as-cast CrFeCoNiNb_x_Mo_1−x_ alloys in 1 M deaerated H_2_SO_4_ solution at 30 °C. The potentiodynamic polarization data of these polarization curves are listed in Table 7. The corrosion potential (*E*_corr_) and the corrosion current densities (*i*_corr_) of CrFeCoNiNb_x_Mo_1−x_ alloys were very close; the CrFeCoNiNb_0.25_Mo_0.75_ alloy had the lowest *i*_corr_ of about 5 μA/cm^2^. The potentiodynamic polarization curve of CrFeCoNiNb_0.25_Mo_0.75_ and CrFeCoNiNb_0.75_Mo_0.25_ alloys had apparent anodic peaks, but the CrFeCoNiNb_0.5_Mo_0.5_ alloy had no anodic peak. The passivation potential (*E*_pp_) and critical current density (*i*_crit_) of the anodic peaks of the alloys are listed in Table 7. This indicated that the CrFeCoNiNb_0.5_Mo_0.5_ alloys easily entered the passivation regions and formed the passive films during corrosion in H_2_SO_4_ solution among these alloys. The current densities of the passivation regions (*i*_pass_) of CrFeCoNiNb_0.5_Mo_0.5_ and CrFeCoNiNb_0.75_Mo_0.25_ alloys were about 12 A/cm^2^, but the CrFeCoNiNb_0.25_Mo_0.75_ alloy had a larger passivation current density (*i*_pass_). The passivation regions of these alloys were all breakdown at a potential (*E*_b_) of about 1.2 V (SHE) because of oxygen evolution reaction [26]. In order to compare the properties of corrosion potential, corrosion current density, anodic peak and passivation region of each alloy in the present study, the CrFeCoNiNb_0.5_Mo_0.5_ alloy had the best corrosion resistance among the alloys in H_2_SO_4_ solution.

Figure 10 shows the potentiodynamic polarization curves of the as-cast CrFeCoNiNb_x_Mo_1−x_ alloys in 1 M deaerated NaCl solution at 30 °C. The potentiodynamic polarization data of these polarization curves are listed in Table 8. The cathodic limit current density (*i*_L_) was found in these CrFeCoNiNb_x_Mo_1−x_ alloys. The cathodic limit current density meant that the maximum reaction rate was limited because of the diffusion rate of hydroxyl ions (OH^−^) in the solution [26]. The corrosion potential (*E*_corr_) of CrFeCoNiNb_0.25_Mo_0.75_ alloy was more positive than the other alloys; the corrosion potential (*E*_corr_) of CrFeCoNiNb_0.5_Mo_0.5_ and CrFeCoNiNb_0.75_Mo_0.25_ alloy was very close. Moreover, the corrosion current density (*i*_corr_) of CrFeCoNiNb_0.25_Mo_0.75_ alloy was larger than the other alloys. The corrosion current density (*i*_corr_) of CrFeCoNiNb_0.5_Mo_0.5_ and CrFeCoNiNb_0.75_Mo_0.25_ alloys were about 10 μA/cm^2^. All of the CrFeCoNiNb_x_Mo_1−x_ alloys had apparent anodic peaks in 1 M deaerated NaCl solution. The passivation regions of CrFeCoNiNb_0.5_Mo_0.5_ and CrFeCoNiNb_0.75_Mo_0.25_ alloys were better than that of CrFeCoNiNb_0.25_Mo_0.75_ alloy. The minimum passivation current densities (*i*_pass_) of CrFeCoNiNb_0.5_Mo_0.5_ and CrFeCoNiNb_0.75_Mo_0.25_ alloys were about 15 μA/cm^2^. When comparing the properties of corrosion potential, corrosion current density, anodic peak and passivation region of each alloy in the present study, the CrFeCoNiNb_0.5_Mo_0.5_ alloy had the best corrosion resistance among the alloys in NaCl solution.

The corrosion rate of the alloys in deaerated 1 M H_2_SO_4_ and 1 M NaCl solutions can be calculated by assuming that all of the corrosion types of the alloys in these solutions are a type of uniform corrosion. Therefore, the relationship between corrosion depth of one year and corrosion current density is listed as the following Equation [27]:(1)$$\mathrm{corrosion}\mathrm{depth}=\frac{M\cdot i_{\mathrm{corr}}\cdot t}{\rho \cdot n\cdot F}$$
where *M* is the average atomic mass (g/mol), *i*_corr_ is the corrosion current density (A/cm^2^), *t* is the corrosion time (31,536,000 s, 1 yr), *ρ* is the average density (g/cm^3^), *n* is the number of average valence electron and *F* is the Faraday constant (96,500 C/mol). This study assumes that the average density of an alloy is *ρ* = ∑X_i_*ρ*_i_, where X_i_ and *ρ*_i_ are the molar fraction and density of element i. The corrosion rates (mm per year) of the alloys in deaerated 1 M H_2_SO_4_ and 1 M NaCl solutions are listed in Table 9. The CrFeCoNiNb_0.3_Mo_0.3_ alloy had the smallest corrosion rate (0.0732 mm/yr) in 1 M deaerated H_2_SO_4_ solution, and alloy CrFeCoNiNb_0.75_Mo_0.25_ had the largest corrosion rate (0.3315 mm/yr) in this solution. The CrFeCoNiNb_0.15_Mo_0.15_ alloy had the smallest corrosion rate (0.0425 mm/yr) in 1 M deaerated NaCl solution, and CrFeCoNiNb_0.5_Mo_0.5_ and CrFeCoNiNb_0.75_Mo_0.25_ alloys had larger corrosion rate in this solution. The corrosion rate of CrFeCoNiNb_0.5_Mo_0.5_ alloy (0.1152 mm/yr) in 1 M deaerated NaCl solution was higher than that of CrFeCoNiNb_0.15_Mo_0.15_ alloy in the same solution. However, the CrFeCoNiNb_0.5_Mo_0.5_ alloy still possessed the best combination of corrosion resistance and hardness among these alloys.

## 4. Conclusions

The microstructures, hardness and corrosion properties of hypoeutectic CrFeCoNiNb_x_Mo_x_ alloys (x = 0.15 and 0.3) and hypereutectic CrFeCoNiNb_x_Mo_1−x_ alloys (x = 0.25, 0.5 and 0.75) were studied. There were two phases (FCC and HCP) in these alloys. The dendrites of hypoeutectic CrFeCoNiNb_x_Mo_x_ alloys (x = 0.15 and 0.3) were an FCC phase; the interdendrities of these alloys were a eutectic structure which the phases were HCP and FCC phases. The dendrites of hypereutectic CrFeCoNiNb_x_Mo_1−x_ alloys (x = 0.25, 0.5 and 0.75) were an HCP-structured laves phase, and the interdendrities of these alloys were a eutectic structure with HCP and FCC phases. Increasing the content of Nb and Mo would increase the hardness of the alloys because of the formation of the hard HCP phase and the effect of solid solution strengthening. The CrFeCoNiNb_0.75_Mo_0.25_ alloy had the highest hardness of 625 HV among the alloys in present study. After potentiodynamic polarization test in deaerated 1 M H_2_SO_4_ and 1 M NaCl solutions at 30 °C, the CrFeCoNiNb_0.5_Mo_0.5_ alloy had the best corrosion resistance among these alloys by comparing the properties of corrosion potential, corrosion current density, anodic peak and passivation region of the alloys. Therefore, CrFeCoNiNb_0.5_Mo_0.5_ alloy was the best alloy among these alloys by comparing the corrosion properties and hardness.

## Figures and Tables

**Figure 1 materials-15-02262-f001:**
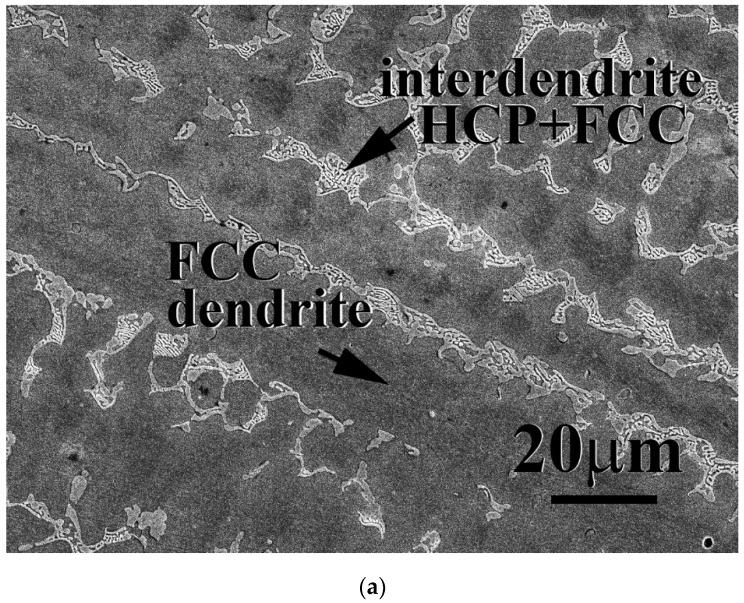
SEM micrographs of as-cast CrFeCoNiNb_x_Mo_x_ alloys: (**a**) CrFeCoNiNb_0.15_Mo_0.15_ alloy; (**b**) CrFeCoNiNb_0.3_Mo_0.3_ alloy; (**c**) CrFeCoNiNb_0.5_Mo_0.5_ alloy.

**Figure 2 materials-15-02262-f002:**
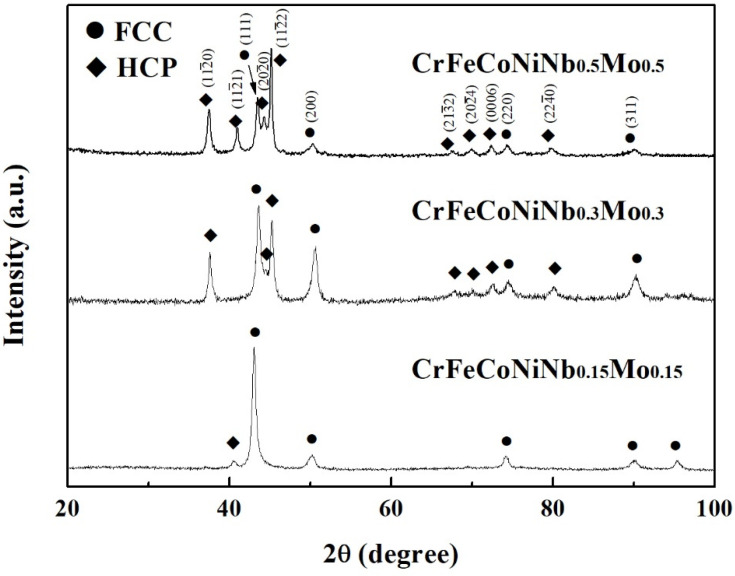
XRD patterns of as-cast CrFeCoNiNb_x_Mo_x_ alloys.

**Figure 3 materials-15-02262-f003:**
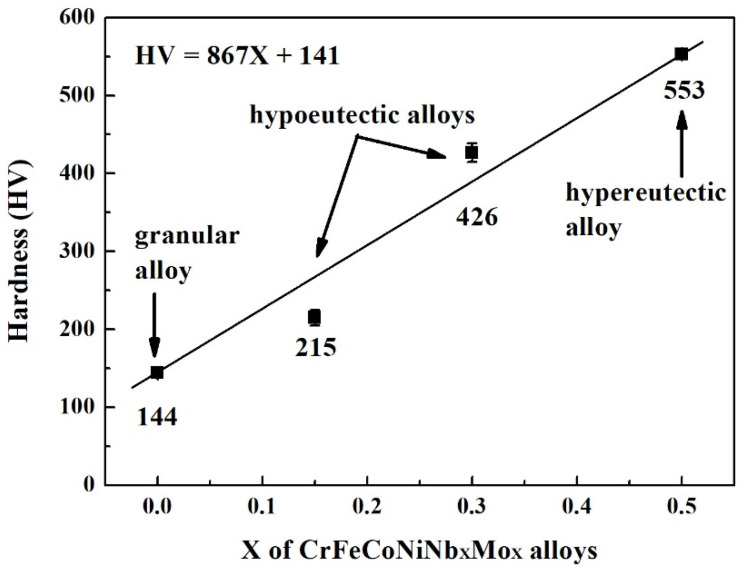
Hardness of as-cast CrFeCoNiNb_x_Mo_x_ alloys. Each value is the average hardness of the alloy.

**Figure 4 materials-15-02262-f004:**
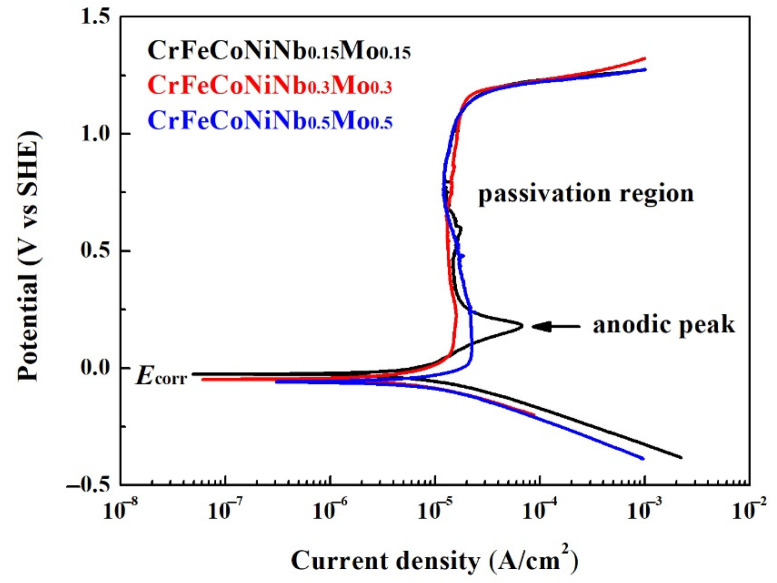
Potentiodynamic polarization curves of as-cast CrFeCoNiNb_x_Mo_x_ alloys tested in the 1 M deaerated sulfuric acid solution at 30 °C.

**Figure 5 materials-15-02262-f005:**
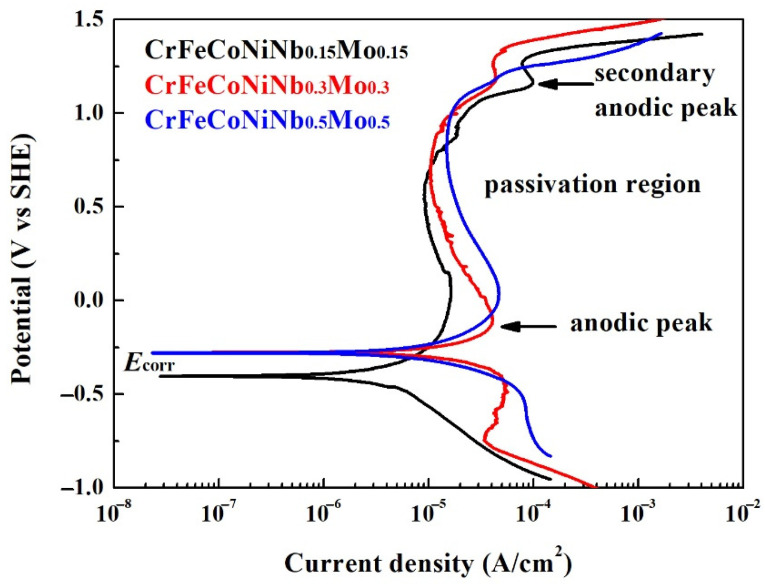
Potentiodynamic polarization curves of as-cast CrFeCoNiNb_x_Mo_x_ alloys tested in the 1 M deaerated sodium chloric solution at 30 °C.

**Figure 6 materials-15-02262-f006:**
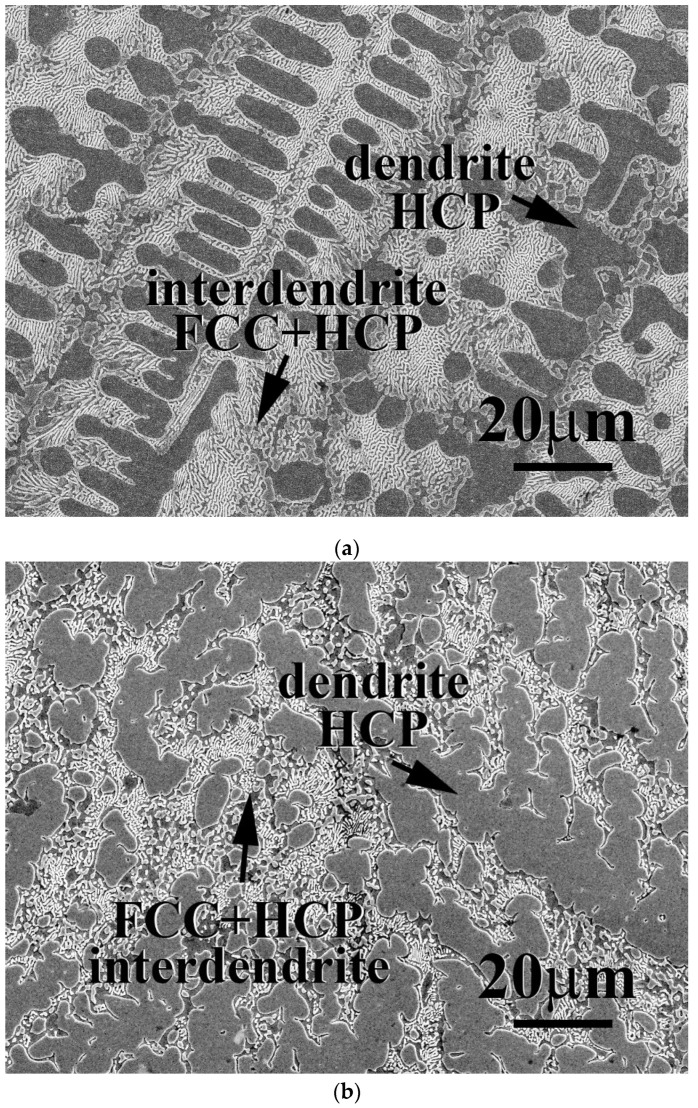
SEM micrographs of as-cast CrFeCoNiNb_x_Mo_1−x_ alloys: (**a**) CrFeCoNiNb_0.25_Mo_0.75_ alloy and (**b**) CrFeCoNiNb_0.75_Mo_0.25_ alloy.

**Figure 7 materials-15-02262-f007:**
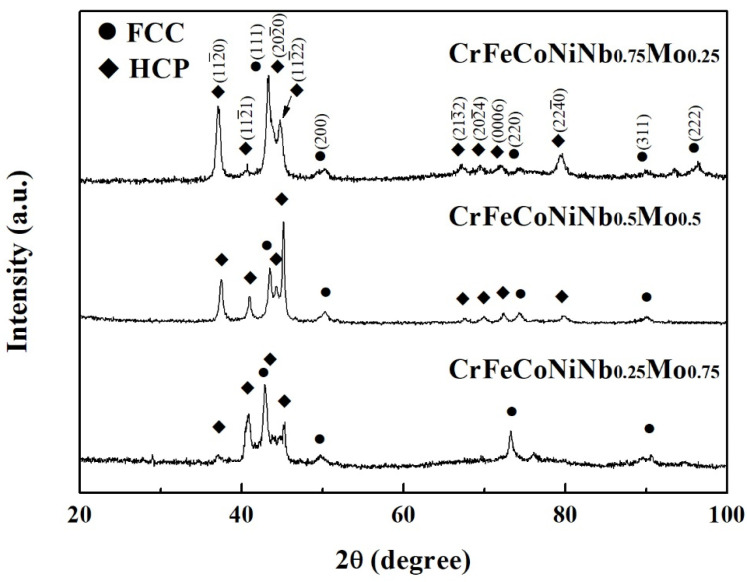
XRD patterns of as-cast CrFeCoNiNb_x_Mo_1−x_ alloys.

**Figure 8 materials-15-02262-f008:**
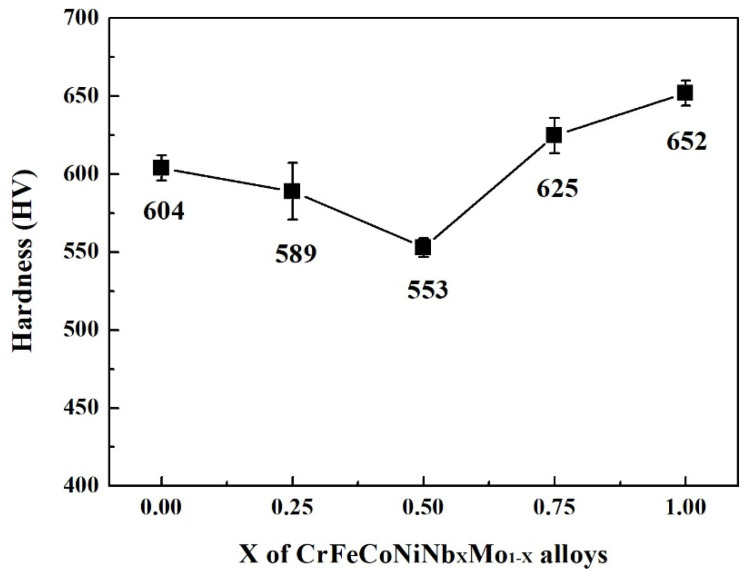
Hardness of as-cast CrFeCoNiNb_x_Mo_1−x_ alloys. Each value is the average hardness of the alloy. Some of the data are from our previous study [23].

**Figure 9 materials-15-02262-f009:**
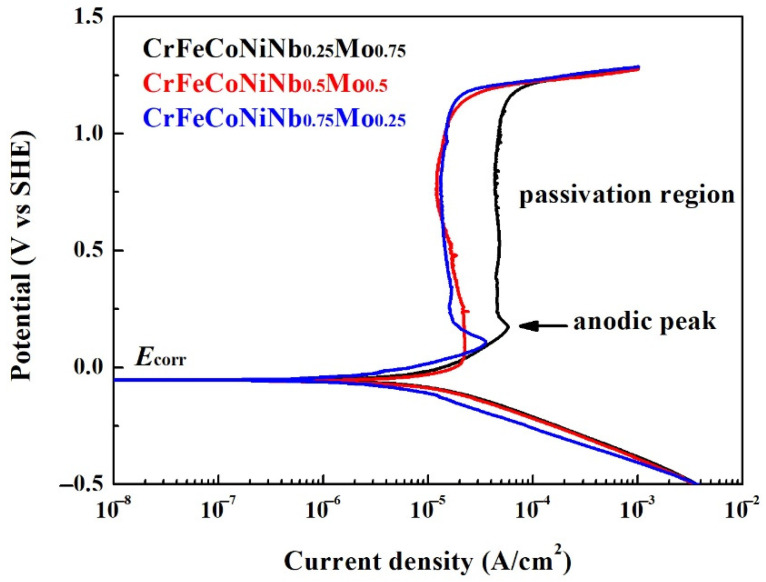
Potentiodynamic polarization curves of as-cast CrFeCoNiNb_x_Mo_1−x_ alloys tested in the 1 M deaerated sulfuric acid solution at 30 °C.

**Figure 10 materials-15-02262-f010:**
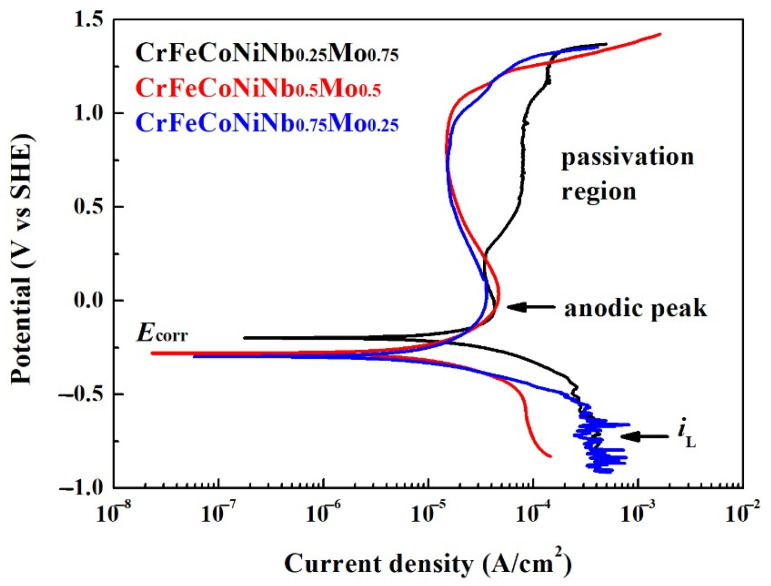
Potentiodynamic polarization curves of as-cast CrFeCoNiNb_x_Mo_1−x_ alloys tested in the 1 M deaerated sodium chloric solution at 30 °C.

**Table 1 materials-15-02262-t001:** The nominal compositions of the as-cast alloys.

Alloys	Weight Percent					
	Cr	Fe	Co	Ni	Nb	Mo
CrFeCoNiNb_0.15_Mo_0.15_	20.49	22.00	23.22	23.13	5.49	5.67
CrFeCoNiNb_0.3_Mo_0.3_	18.43	19.79	20.89	20.81	9.88	10.20
CrFeCoNiNb_0.5_Mo_0.5_	16.25	17.46	18.42	18.35	14.52	14.99
CrFeCoNiNb_0.25_Mo_0.75_	16.22	17.42	18.38	18.31	7.24	22.44
CrFeCoNiNb_0.75_Mo_0.25_	16.29	17.50	18.46	18.40	21.83	7.52

**Table 2 materials-15-02262-t002:** The chemical compositions of the as-cast CrFeCoNiNb_x_Mo_x_ alloys and the phases in each alloy analyzed by SEM/EDS.

Alloys	Weight Percent					
	Cr	Fe	Co	Ni	Nb	Mo
CrFeCoNiNb_0.15_Mo_0.15_						
overall	23.9 ± 0.9	24.2 ± 4.9	25.6 ± 0.4	17.2 ± 4.8	5.0 ± 0.2	4.1 ± 0.2
FCC	24.9 ± 1.3	25.3 ± 0.3	22.3 ± 1.1	20.5 ± 1.2	3.1 ± 0.7	3.9 ± 0.6
HCP	19.8 ± 0.5	16.0 ± 1.4	26.6 ± 1.3	16.6 ± 0.5	10.4 ± 0.3	10.6 ± 0.2
CrFeCoNiNb_0.3_Mo_0.3_						
overall	23.1 ± 1.3	20.4 ± 1.5	18.1 ± 2.1	21.7 ± 1.6	9.2 ± 0.5	7.5 ± 0.8
FCC	23.6 ± 2.0	23.3 ± 2.2	20.4 ± 2.8	25.8 ± 3.0	2.0 ± 0.4	5.1 ± 0.6
HCP	18.7 ± 0.6	14.6 ± 0.2	17.9 ± 2.6	17.8 ± 0.2	19.2 ± 0.2	11.8 ± 2.5
CrFeCoNiNb_0.5_Mo_0.5_						
overall	21.4 ± 0.4	19.4 ± 1.6	19.8 ± 0.3	17.6 ± 2.1	11.0 ± 0.2	10.8 ± 0.7
FCC	22.1 ± 0.9	20.4 ± 1.4	18.4 ± 0.3	22.8 ± 0.4	7.1 ± 1.2	9.2 ± 0.5
HCP	17.3 ± 0.3	23.6 ± 5.2	19.1 ± 0.8	14.2 ± 0.8	16.9 ± 0.1	15.6 ± 0.9

**Table 3 materials-15-02262-t003:** Potentiodynamic polarization data of the as-cast CrFeCoNiNb_x_Mo_x_ alloys in 1 M deaerated H_2_SO_4_ solution at 30 °C.

	CrFeCoNiNb_0.15_Mo_0.15_	CrFeCoNiNb_0.3_Mo_0.3_	CrFeCoNiNb_0.5_Mo_0.5_
*E*_corr_ (V vs. SHE)	−0.03	−0.05	−0.06
*i*_corr_ (μA/cm^2^)	11.9	7.0	10.0
*E*_pp_ (V vs. SHE)	0.18	N/A	N/A
*i*_crit_ (μA/cm^2^)	67.5	N/A	N/A
*i*_pass_ (μA/cm^2^)	12.5	13.1	12.2
*E*_b_ (V vs. SHE)	1.21	1.19	1.19

**Table 4 materials-15-02262-t004:** Standard electrode potential at 25 °C [25].

Reaction	Electrode Potential (*E*° vs. SSE)
Cr, Cr^3+^	−0.74
Fe, Fe^2+^	−0.44
Co, Co^2+^	−0.227
Ni, Ni^2+^	−0.25
Nb, Nb^3+^	−1.10
Mo, Mo^3+^	−0.20

**Table 5 materials-15-02262-t005:** Potentiodynamic polarization data of the as-cast CrFeCoNiNb_x_Mo_x_ alloys in 1 M deaerated NaCl solution at 30 °C.

	CrFeCoNiNb_0.15_Mo_0.15_	CrFeCoNiNb_0.3_Mo_0.3_	CrFeCoNiNb_0.5_Mo_0.5_
*E*_corr_ (V vs. SHE)	−0.40	−0.28	−0.28
*i*_corr_ (μA/cm^2^)	4.2	10.0	10.7
*E*_pp_ (V vs. SHE)	0.13	−0.10	0.03
*i*_crit_ (μA/cm^2^)	16.2	40.9	47.0
*i*_pass_ (μA/cm^2^)	9.1	10.5	15.0
*E*_b_ (V vs. SHE)	1.31	1.33	1.19

**Table 6 materials-15-02262-t006:** The chemical compositions of the as-cast CrFeCoNiNb_x_Mo_1−x_ alloys and the phases in each alloy analyzed by SEM/EDS.

Alloys	Weight Percent					
	Cr	Fe	Co	Ni	Nb	Mo
CrFeCoNiNb_0.25_Mo_0.75_						
overall	16.3 ± 3.4	17.6 ± 1.6	18.5 ± 1.0	18.1 ± 1.6	7.8 ± 1.3	21.7 ± 2.5
FCC	21.7 ± 2.2	21.2 ± 0.5	18.1 ± 1.9	25.6 ± 2.4	4.4 ± 3.3	9.0 ± 0.1
HCP	14.8 ± 2.3	15.2 ± 2.8	20.2 ± 1.5	15.4 ± 1.3	8.7 ± 0.2	25.7 ± 0.6
CrFeCoNiNb_0.75_Mo_0.25_						
overall	16.9 ± 0.6	17.5 ± 2.6	18.8 ± 2.0	18.9 ± 1.2	20.2 ± 0.2	7.7 ± 0.2
FCC	23.2 ± 2.5	23.8 ± 1.0	19.3 ± 0.6	27.7 ± 2.7	3.6 ± 0.1	2.4 ± 1.7
HCP	14.4 ± 0.6	15.5 ± 1.4	21.0 ± 0.4	14.1 ± 0.7	26.5 ± 1.1	8.5 ± 0.5

**Table 7 materials-15-02262-t007:** Potentiodynamic polarization data of the as-cast CrFeCoNiNb_x_Mo_1−x_ alloys in 1 M deaerated H_2_SO_4_ solution at 30 °C.

	CrFeCoNiNb_0.25_Mo_0.75_	CrFeCoNiNb_0.5_Mo_0.5_	CrFeCoNiNb_0.75_Mo_0.25_
*E*_corr_ (V vs. SHE)	−0.06	−0.06	−0.06
*i*_corr_ (μA/cm^2^)	10.2	10.0	4.9
*E*_pp_ (V vs. SHE)	0.17	N/A	0.11
*i*_crit_ (μA/cm^2^)	58.3	N/A	35.7
*i*_pass_ (μA/cm^2^)	44.1	12.2	11.9
*E*_b_ (V vs. SHE)	1.21	1.19	1.19

**Table 8 materials-15-02262-t008:** Potentiodynamic polarization data of the as-cast CrFeCoNiNb_x_Mo_1−x_ alloys in 1 M deaerated NaCl solution at 30 °C.

	CrFeCoNiNb_0.25_Mo_0.75_	CrFeCoNiNb_0.5_Mo_0.5_	CrFeCoNiNb_0.75_Mo_0.25_
*E*_corr_ (V vs. SHE)	−0.20	−0.28	−0.30
*i*_corr_ (μA/cm^2^)	31.0	10.7	10.8
*E*_pp_ (V vs. SHE)	−0.04	0.03	0.03
*i*_crit_ (μA/cm^2^)	43.2	47.0	36.0
*i*_pass_ (μA/cm^2^)	34.4	15.0	15.6
*E*_b_ (V vs. SHE)	1.32	1.19	1.26

**Table 9 materials-15-02262-t009:** Corrosion rates of the as-cast alloys in deaerated 1 M H_2_SO_4_ and 1 M NaCl solutions at 30 °C.

Alloys	1 M H_2_SO_4_ (mm/yr)	1 M NaCl (mm/yr)
CrFeCoNiNb_0.15_Mo_0.15_	0.1204	0.0425
CrFeCoNiNb_0.3_Mo_0.3_	0.0732	0.1043
CrFeCoNiNb_0.25_Mo_0.75_	0.1091	0.0532
CrFeCoNiNb_0.5_Mo_0.5_	0.1077	0.1152
CrFeCoNiNb_0.75_Mo_0.25_	0.3315	0.1172

## Data Availability

Not applicable.

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
