# Peer review of "The Effects of Niobium and Molybdenum on the Microstructures and Corrosion Properties of CrFeCoNiNbxMoy Alloys"

_materials, 2022, doi:10.3390/ma15062262_

Round 1

Reviewer 1 Report

1/ English must be significantly improved throughout the whole manuscript.

2/ The last paragraph of Introduction is a little bit weird. It is rather something like Abstract. I am not sure if this is the best way. Reconsider this part of Introduction.

3/ Novelty and aim of the study must be clearly emphasized.

4/ Experimental. The authors mention only the melting conditions and the methodology of corrosion tests. There is nothing on materials preparing, I mean: hot working / cold working / plastic deformation / heat treatment, etc. The data is crucial for further microstructure and corrosion-related results.

5/ Figure 1. The FCC and HCP is not enough clear in the figure. Rather, a type of the phase should be indicated.

6/ The same concerns the X-ray data, which should identify a type of the phase. The planes should be given in X-ray patterns.

7/ Without clear identification of the phases the report and its conclusions are of low scientific value.

8/ There is no discussion of the obtained data, which means that ghe study is only a technical report, not a scientific paper.

Author Response

1/ English must be significantly improved throughout the whole manuscript.

Reply: We have modified our manuscript.

2/ The last paragraph of Introduction is a little bit weird. It is rather something like Abstract. I am not sure if this is the best way. Reconsider this part of Introduction.

Reply: We have modified this paragraph. This paragraph described the reasons why we studied these alloys.

3/ Novelty and aim of the study must be clearly emphasized.

Reply: We have modified the manuscript.

4/ Experimental. The authors mention only the melting conditions and the methodology of corrosion tests. There is nothing on materials preparing, I mean: hot working / cold working / plastic deformation / heat treatment, etc. The data is crucial for further microstructure and corrosion-related results.

Reply: We used the as-cast alloys, no other conditions.

5/ Figure 1. The FCC and HCP is not enough clear in the figure. Rather, a type of the phase should be indicated.

Reply: We have enlarged and modified the images.

6/ The same concerns the X-ray data, which should identify a type of the phase. The planes should be given in X-ray patterns.

Reply: We have modified the X-ray data.

7/ Without clear identification of the phases the report and its conclusions are of low scientific value.

Reply: We used TEM to analyze the structures of the phases in the alloys. The granular structure of CrFeCoNi was published in Mater. Chem. Phys. 2017, vol.186, 534-540. The phases of the dendritic structures in the CrFeCoNiNb and CrFeCoNiNb0.5Mo0.5 alloys were published in Entropy 2018, 20, 648. The TEM images and the corresponding diffraction patterns were shown in the articles. Therefore, we can confirm the structures of the alloys; and this study did not show the images again. We mentioned this and listed the references in the first paragraph in section 3.1. Attachement shows the TEM images of the alloys.

8/ There is no discussion of the obtained data, which means that ghe study is only a technical report, not a scientific paper.

Reply: We have modified the manuscript. We discussed the corrosion rates of the alloys from the corrosion current densities of the alloys, as shown in the last paragraph in “Results and discussion”.

Reviewer 2 Report

How can one conclude the HCP, FCC structure from the SEM micrograph? It is misguiding. SAED or EBSD etc. can give some idea. 

What is the accuracy of the EDS? How the uncertainty is calculated. 

XRD plots are not conclusive. Some of the peaks are missing and then appear in other samples. Please try Reitveld fittings. Also, provide the standard JCPDS of both phases (at the bottom of the plot) so that one can compare the XRD pattern. 

Author Response

How can one conclude the HCP, FCC structure from the SEM micrograph? It is misguiding. SAED or EBSD etc. can give some idea. 

Reply: We used TEM to analyze the structures of the phases in the alloys. The granular structure of CrFeCoNi was published in Mater. Chem. Phys. 2017, vol.186, 534-540. The phases of the dendritic structures in the CrFeCoNiNb and CrFeCoNiNb0.5Mo0.5 alloys were published in Entropy 2018, 20, 648. The TEM images and the corresponding diffraction patterns were shown in the articles. Therefore, we can confirm the structures of the alloys; and this study did not show the images again. We mentioned this and listed the references in the first paragraph in section 3.1. Attachement shows the TEM images of the alloys.

What is the accuracy of the EDS? How the uncertainty is calculated. 

Reply: The data of EDS were done by the operator. We measured five point (at least) to average the compositions.

XRD plots are not conclusive. Some of the peaks are missing and then appear in other samples. Please try Reitveld fittings. Also, provide the standard JCPDS of both phases (at the bottom of the plot) so that one can compare the XRD pattern. 

Reply: As described above, we did not used the JCPDS data. We identified the phases from our previous TEM observation.

Reviewer 3 Report

The article is devoted to the study of the effect of niobium and molybdenum on the microstructure and corrosion properties of high-entropy alloys CrFeCoNiNbxMox and CrFeCoNiNbxMo1-x, which were obtained by arc melting in an argon atmosphere. These studies are of scientific interest and practical significance, since the selected compositions have a high potential for practical application in modern materials science. However, before the article is accepted for publication, the authors should make changes and answer the questions posed.
1. In the introduction and abstract, the authors should write in more detail the purpose and relevance of this work, taking into account the latest achievements in this field. Also, the article requires significant revision, since in this form it looks more like an analytical report than an article.
2. The presence of grains, according to the presented images, requires additional explanations, is it related to the formation of individual phase inclusions?
3. The authors should provide more details when describing the results of the elemental content of the alloy components, as well as the dependences associated with their change.
4. Due to which the increase in hardness in alloys with the addition of niobium and molybdenum is due, as a rule, doping leads to a complex crystal structure and an increase in dislocation density.
5. X-ray data require additional explanations, as well as the presentation of the results of the phase composition depending on the concentration of the dopant components.

Author Response

  1. In the introduction and abstract, the authors should write in more detail the purpose and relevance of this work, taking into account the latest achievements in this field. Also, the article requires significant revision, since in this form it looks more like an analytical report than an article.
    Reply: We have modified our manuscript.
  2. The presence of grains, according to the presented images, requires additional explanations, is it related to the formation of individual phase inclusions?
    Reply: We have modified the figures.
  3. The authors should provide more details when describing the results of the elemental content of the alloy components, as well as the dependences associated with their change.
    Reply: The compositions of the phases in each alloy are listed in Table 2 and Table 6. The element content of the alloys are described in the text.
  4. Due to which the increase in hardness in alloys with the addition of niobium and molybdenum is due, as a rule, doping leads to a complex crystal structure and an increase in dislocation density.
    Reply: We have modified the manuscript, Line 147-150.
  5. X-ray data require additional explanations, as well as the presentation of the results of the phase composition depending on the concentration of the dopant components.

Reply: We have modified the manuscript.

Reviewer 4 Report

The paper ”The effects of niobium and molybdenum on the microstructures and corrosion properties of CrFeCoNiNbxMoy alloys” is suitable for publication just after some minor revisions:

-authors should add the ICDD files for XRD analysis

- authors should add corrosion rate in table 5,7 and 8 and give explanations in text.

Author Response

-authors should add the ICDD files for XRD analysis

Reply: We used TEM to analyze the structures of the phases in the alloys. The granular structure of CrFeCoNi was published in Mater. Chem. Phys. 2017, vol.186, 534-540. The phases of the dendritic structures in the CrFeCoNiNb and CrFeCoNiNb0.5Mo0.5 alloys were published in Entropy 2018, 20, 648. The TEM images and the corresponding diffraction patterns were shown in the articles. Therefore, we can confirm the structures of the alloys; and this study did not show the images again. We did not use the ICDD files for XRD analysis. We mentioned this and listed the references in the first paragraph in section 3.1. Attachement shows the TEM images of the alloys.

- authors should add corrosion rate in table 5,7 and 8 and give explanations in text.

Reply: The corrosion rates of the alloys were included in the manuscript, the last paragraph in the “Results and discussion”.

Round 2

Reviewer 1 Report

The authors addressed all my comments. The answers and provided modifications are enough to be accepted. The value of the manuscript is improved.

Author Response

Thank you for your valuable commands.

Reviewer 2 Report

Please mark the XRD peaks and see if some other phases are formed or not. XRD peaks have to be indexed using the ICDD/JCPDS card and not from the TEM diffraction pattern. WE index even the SAED using the JCPDS card. 

To say that operator has done the EDS and you have no idea is not a scientific answer. I mean the EDS accuracy is not good enough

Author Response

Please mark the XRD peaks and see if some other phases are formed or not. XRD peaks have to be indexed using the ICDD/JCPDS card and not from the TEM diffraction pattern. WE index even the SAED using the JCPDS card. 

Reply: The angles of the XRD peaks will change with the compositions of the phases in alloys. So we used TEM DPs to make sure the peaks of XRD patterns. The TEM DPs of FCC and HCP phases are easily identified. We always used this method.

To say that operator has done the EDS and you have no idea is not a scientific answer. I mean the EDS accuracy is not good enough

Reply: EDS is not an accuracy equipment. But it is easy to examine. We detected the compositions of the phases, so we could understand that the phase was HCP or FCC phase. Because HCP phase had higher Nb- and Mo-content. We measured five times (at least) to average the compositions of a phase, was also caused by the accuracy of EDS.

Reviewer 3 Report

The authors answered all the questions posed. The article may be accepted for publication.

Author Response

Thank you for your valuable commands.